# Syndecan-1 Overexpressing Mesothelioma Cells Inhibit Proliferation, Wound Healing, and Tube Formation of Endothelial Cells

**DOI:** 10.3390/cancers13040655

**Published:** 2021-02-06

**Authors:** Joman Javadi, Ghazal Heidari-Hamedani, Angelika Schmalzl, Tünde Szatmári, Muzaffer Metintas, Pontus Aspenström, Anders Hjerpe, Katalin Dobra

**Affiliations:** 1Department of Laboratory Medicine, Division of Pathology, Karolinska Institutet, SE-14157 Stockholm, Sweden; joman.javadi@ki.se (J.J.); heidari.ghazal@gmail.com (G.H.-H.); Angelika.Schmalzl@uk-erlangen.de (A.S.); tszatmari@yahoo.com (T.S.); anders.hjerpe@ki.se (A.H.); 2Department of Chest, Eskisehir Osmangazi University Medical Faculty, 26040 Eskisehir, Turkey; mmetintas@ogu.edu.tr; 3Department of Immunology, Genetics and Pathology, Uppsala University Rubeck Laboratory, SE-75185 Uppsala, Sweden; pontus.aspenstrom@igp.uu.se

**Keywords:** angiogenesis, endothelial cells, mesothelioma, syndecan-1, tubulogenesis

## Abstract

**Simple Summary:**

The transmembrane proteoglycan syndecan-1 (SDC-1) is an important mediator of cell-matrix interactions. The heparan sulfate side-chains of SDC-1 can bind to a multitude of growth factors, cytokines, and chemokines, thereby regulating a plethora of physiological and pathological processes, including angiogenesis. The extracellular region of SDC-1 can be released from the cell surface by the action of sheddases including matrix metalloproteinase-7 and 9, resulting in a soluble protein that is still active and can act as a competitive activator or inhibitor of the cell surface receptor. Accelerated shedding and loss of cell surface SDC-1 is associated with epithelial to mesenchymal transition (EMT) and achievement of a more invasive phenotype in malignant mesothelioma (MM). Transfection with SDC-1 reverts the morphology in epithelioid direction and inhibits the proliferation and migration of MM cells. This study aimed to investigate the role of SDC-1 in angiogenesis. We demonstrate that overexpression and silencing of SDC-1 alters the secretion of angiogenic proteins in MM cells. Upon SDC-1 overexpression, several factors collectively inhibit the proliferation, wound closure, and tube formation of endothelial cells, whereas SDC-1 silencing only affects wound healing.

**Abstract:**

Malignant mesothelioma (MM) is an aggressive tumor of the serosal cavities. Angiogenesis is important for mesothelioma progression, but so far, anti-angiogenic agents have not improved patient survival. Our hypothesis is that better understanding of the regulation of angiogenesis in this tumor would largely improve the success of such a therapy. Syndecan-1 (SDC-1) is a transmembrane heparan sulfate proteoglycan that acts as a co-receptor in various cellular processes including angiogenesis. In MM, the expression of SDC-1 is generally low but when present, SDC-1 associates to epithelioid differentiation, inhibition of tumor cell migration and favorable prognosis, meanwhile SDC-1 decrease deteriorates the prognosis. In the present study, we studied the effect of SDC-1 overexpression and silencing on MM cells ability to secrete angiogenic factors and monitored the downstream effect of SDC-1 modulation on endothelial cells proliferation, wound healing, and tube formation. This was done by adding conditioned medium from SDC-1 transfected and SDC-1 silenced mesothelioma cells to endothelial cells. Moreover, we investigated the interplay and molecular functional changes in angiogenesis in a co-culture system and characterized the soluble angiogenesis-related factors secreted to the conditioned media. We demonstrated that SDC-1 over-expression inhibited the proliferation, wound healing, and tube formation of endothelial cells. This effect was mediated by a multitude of angiogenic factors comprising angiopoietin-1 (Fold change ± SD: 0.65 ± 0.07), FGF-4 (1.45 ± 0.04), HGF (1.33 ± 0.07), NRG1-β1 (1.35 ± 0.08), TSP-1 (0.8 ± 0.02), TIMP-1 (0.89 ± 0.01) and TGF-β1 (1.35 ± 0.01). SDC-1 silencing increased IL8 (1.33 ± 0.06), promoted wound closure, but did not influence the tube formation of endothelial cells. Pleural effusions from mesothelioma patients showed that Vascular Endothelial Growth Factor (VEGF) levels correlate to soluble SDC-1 levels and have prognostic value. In conclusion, SDC-1 over-expression affects the angiogenic factor secretion of mesothelioma cells and thereby inhibits endothelial cells proliferation, tube formation, and wound healing. VEGF could be used in prognostic evaluation of mesothelioma patients together with SDC-1.

## 1. Introduction

Malignant mesothelioma (MM) is the aggressive primary tumor of the serosal cavities associated, in the majority of cases, to asbestos exposure [1,2]. The most common form of the disease affects the pleura, followed by the peritoneum and pericardium. The tumor shows poor prognosis with only modest improvement by the available treatment regimens [3].

Tumor vascularization is a prerequisite for tumor growth and metastasis [4]. Pro- and anti-angiogenic proteins are produced during angiogenesis to enhance or inhibit the process when it is needed [5]. Angiogenesis is crucial in mesothelioma progression as high microvascular density and angiogenesis-related proteins have been associated to adverse prognosis and advanced disease [6,7,8,9]. Therefore, there is a strong rationale for using anti-angiogenic agents for the treatment of MM; highlighting the further need of novel treatment options [10,11] along with better understanding of the underlying mechanisms for angiogenic signaling.

Syndecan-1 (SDC-1) is a transmembrane heparan sulfate proteoglycan that maintains the epithelioid differentiation of MM cells, promotes their adhesion, and inhibits their migration, proliferation [12], and angiogenesis [13]. SDC-1 acts as a co-receptor for a wide range of extracellular ligands, including VEGF, a key signaling molecule in angiogenesis both during embryogenesis and tumor growth [5,14]. Membrane-bound SDC-1 regulates angiogenesis via interaction and activation of αvβ3 and αvβ5 integrins and insulin-like growth factor-1receptor (IGF-1R), forming a ternary receptor complex. This complex activates intracellular integrin signaling and is present mainly on tumor cells and activated endothelial cells during angiogenesis. At the same time, the responsible peptide sequence for αvβ3 and αvβ5 integrins and IGF-1R interaction on SDC1 can be mimicked by a short peptide known as synstatin. This peptide competes with SDC1 and inactivates the integrin/IGF-1R/SDC-1 complex, counteracting the signaling pathways involved in angiogenesis. This angiogenesis modulating potential of SDC-1 may also serve as a therapeutic approach [15].

The extracellular domain of SDC-1 can be shed by proteolytic cleavage of proteases including matrix metalloproteinases (MMPs), membrane-tethered MMPs (MT-MMPs), and A-disintegrin-and-matrix-metalloproteinases (ADAMs) [16]. Shedding generates soluble SDC-1 that can either have similar or antagonistic effects compared to the membrane bound form. Levels of both membrane-bound as well as shed SDC-1 are often altered in malignancies [17]. Cell surface expression of SDC-1 indicates favorable prognosis in malignant mesothelioma [13,18], whereas high levels of shed, soluble SDC-1, correlates to poor prognosis [19].

We have previously shown that upregulation of the SDC-1 hampers mesothelioma cell growth and cell cycle progression and also affects mesenchymal tumor cell adhesion and migration [12,20]. In the current study, we aim to disclose the functional role of SDC-1 in angiogenesis. By over-expressing and silencing SDC-1 in mesothelioma cells, we could elicit changes in a wide range of angiogenesis-related proteins secreted to the conditioned media. These factors collectively influenced angiogenesis by leading to decreased endothelial cell proliferation, wound healing, and tubulogenesis.

## 2. Results

### 2.1. Characterization of the Angiogenesis-Related Proteins Secreted by SDC-1 Overexpressing and SDC-1 Silenced Mesothelioma Cells

To determine the effect of SDC-1 on angiogenesis-related protein secretion of malignant mesothelioma (MM) cells, we analyzed the conditioned medium of SDC-1 overexpressing and SDC-1 silenced cells by Proteome Profiler Array. Modulation of SDC-1 expression elicited different pattern of angiogenesis-related factor secretion, most effect being observed upon SDC-1 overexpression. Seven angiogenesis-related proteins were significantly altered by SDC-1 overexpression including angiopoietin-1 (mean fold change ± SD: 0.65 ± 0.07), fibroblast growth factor-4 (FGF-4) (1.45 ± 0.04), hepatocyte growth factor (HGF) (1.33 ± 0.12), neuroglin-1 beta-1(NRG1-β1) (1.35 ± 0.02), tissue inhibitor of metalloproteinases-1 (TIMP-1) (0.89 ± 0.01), thrombospondin-1 (TSP-1) (0.8 ± 0.04), and transforming growth factor beta-1 (TGF-β1) (1.35 ± 0.02) (Figure 1A), whereas interleukin 8 (IL-8) (1.33 ± 0.06) was increased upon SDC-1 silencing (Figure 1B). Only two genes were significantly affected by both conditions. TSP-1 (1.15 ± 0.17) and TIMP-1 (1.08 ± 0.08) increased when the SDC-1 silenced cells conditioned medium was compared to SDC-1 overexpressing cells; ratios of both conditions being calculated relative to their specific control (*p* < 0.05) (Figure 1C).

### 2.2. Conditioned Medium from SDC-1 Over-Expressing Mesothelioma Cells Inhibits Endothelial Cell Proliferation

Since endothelial cell proliferation is one of the early processes in angiogenesis, we first studied if supernatants from the cells over-expressing SDC-1 may influence endothelial cells proliferation. We observed that endothelial cell proliferation was significantly hampered after incubation with conditioned medium from SDC-1 over-expressing cells compared to Human Umbilical Vein Endothelial Cells (HUVEC)cells growing in conditioned medium from control cells. HUVEC cell number was reduced to 67.2% and 62.6%, respectively, compared to corresponding mock controls after 24 and 48 h incubation with SDC-1 over-expressing conditioned medium (Figure 2). All changes were statistically significant at (* *p* ≤ 0.05).

### 2.3. Conditioned Medium from SDC-1 Overexpressing Mesothelioma Cells Inhibits Endothelial Cell Wound Healing Assay

Next, we tested the paracrine effect of the angiogenic factors secreted by SDC-1 overexpressing MM cells on the wound healing of endothelial cells. This ability is slower in endothelial cells treated with conditioned medium from SDC-1 over-expressing MM cells (−11,541 pixel^2^/h) compared to the corresponding empty vector controls (−17,220 pixel^2^/hour). To further elucidate the involvement of SDC-1 in this process, we added conditioned medium from SDC-1 silenced MM cells to endothelial cells. Wound closure of endothelial cells treated with conditioned medium of SDC-1 silenced cells is faster compared to endothelial cells treated with conditioned medium from scrambled negative control. The wound area closure in HUVEC cells treated with conditioned medium from SDC-1 over-expressing cells was significantly lower compared to cells treated with conditioned medium form control cells. Additionally, the wound area closure in HUVEC cells treated by conditioned medium from SDC-1 silenced cells was significantly higher compared to the cells treated with conditioned medium from scrambled negative control cells (Figure 3A,B).

In addition to proliferation and closure of wound area, the directional movement of endothelial cells towards a gradient of chemo-attractants plays an important role for the formation of new vessel structures during angiogenesis. To analyze how soluble angiogenesis-related factors secreted by SDC-1 overexpressing cells influence chemotaxis of endothelial cells, HUVEC cells were incubated with conditioned medium obtained from SDC-1 over-expressing and mock control cells in combination with VEGF in a trans-well two chamber system.

Chemotactic movement of endothelial cells was estimated from the covered area by migrated cells. Percentage of covered area by HUVEC cells attracted by SDC-1 over-expressing cells conditioned medium was (12.48 ± 5%; mean ± SD), compared to HUVEC cells incubated with conditioned medium collected from mock control cells (8.45 ± 2.5%) (Appendix A).

### 2.4. Conditioned Medium from SDC-1 Over-Expressing Cells Hampers HUVEC Cell Tubulogenesis

Next, we evaluated the functionality of HUVEC cells, by following the formation of capillary-like structures, in presence of conditioned medium from SDC-1 over-expressing cells, SDC-1 silenced cells and their controls. After 6 h incubation of HUVEC cells with conditioned medium from SDC-1 over-expressing cells, the total tube lengths, number of total branching points, and loops were significantly reduced compared to HUVEC cells that were incubated with medium from control cells (Figure 4A,B). Total tube lengths were 12,706 ± 1359 pixel (mean ± SEM) and 6912 ± 678.1 pixel respectively (*p* value = 0.03), total branching points corresponded to 66.89 ± 13.89 and 28.61 ± 6.528 (*p* value = 0.05), and loops were 16.17 ± 1.049 and 4.352 ± 1.591 (*p* value = 0.009) (Figure 5). The other features of tubulogenesis by endothelial cells such as number of total nets and total tubes, mean loop area and perimeter, and percentage of covered area by tubes were also reduced in presence of SDC-1 over-expressing cell conditioned medium compared to the endothelial cells incubated with medium from mock control cells (Appendix A). Silencing of SDC-1 did not inhibit the formation of capillary-like structures of HUVEC cells (Appendix A).

In order to investigate the role of shed SDC-1 on the tube formation of endothelial cells, we silenced MMP-7 in mesothelioma cells. The MMP-7 silencing corresponded to a 44 ± 6% knockdown (*p* < 0.0001) of the target mRNA (Appendix A). We observed that SDC-1 shedding was decreased by nearly 30% in the supernatant silenced for MMP-7 compared to scrambled control (Appendix A). HUVEC cells were then incubated with conditioned medium from MMP7 silenced cells according to described method and tube formation assay was performed. After 6 h incubation of HUVEC cells with conditioned medium from MMP-7 silencing cells, none of the tube formation characteristics were affected compared to the scrambled control (Figure 4C,D and Figure 5).

### 2.5. Shed SDC-1 Is Positively Correlated with VEGF in Pleural Effusions from Mesothelioma Patients

In order to study the relationship between VEGF, as a biomarker of tumor progression and angiogenesis, and shed SDC-1, as a biomarker of malignant effusions [21], we measured the simultaneous expression of the above markers in pleural effusion samples from mesothelioma patients. We found a significantly fair correlation between VEGF and SDC-1 expression in these patients (r = 0.4; *p* < 0.05; n = 39, Figure 6A).

### 2.6. Mesothelioma Patients with Higher Levels of VEGF in Pleural Effusions Displayed Shorter Survival

Patients were divided in two groups, depending on the VEGF concentrations in the effusion. Median survival time of mesothelioma patients with VEGF level <2.125 ng/mL was 14 months (n = 11), compared to 6 months in patients with higher VEGF levels (n = 19; hazard ratio: 2.83, 95% CI: 1.26 to 6.37, *p* = 0.0083) (Figure 6B).

## 3. Discussion

Vascularization is an important process in tumor growth and metastasis [22]. Increased angiogenesis has been reported in mesothelioma, as tumor cells grow [21,23]. Vascular endothelial growth factor (VEGF) is the most prominent angiogenic factor, which is produced by malignant mesothelioma cells [24]. In addition, increased vascularity is reported to be associated with advanced tumor stage and poor prognosis [6,7,8,9]. Therefore, anti-angiogenic therapies may be beneficial in mesothelioma.

In this study, we have shown that conditioned medium from SDC-1 over-expressing mesothelioma cells inhibits the proliferation, wound healing, and tube formation of endothelial cells through a synchronized and collaborative function of multiple pro- and anti-angiogenic factors.

In mesothelioma cells, among the angiogenesis-related proteins, angiopoietin-1, thrombospondin-1 (TSP-1), and tissue inhibitor of metalloproteinases-1 (TIMP-1) were significantly downregulated upon SDC-1 over-expression, whereas IL-8 was upregulated upon SDC-1 silencing. In addition, among proteins which were altered by both SDC-1 over-expression and silencing, TIMP-1 and TSP-1 were significantly upregulated upon SDC-1 silencing.

Mechanistically it has been shown that angiopoietin-1 binds to its receptor Tie2 on endothelial cells and induces sprouting and reorganization of endothelial cells into tubules [25,26] and its inhibition might explain the defects observed in tubule formation, proliferation, and wound closure ability of HUVEC cells. Angiopoietin-1 has been proposed to be a useful prognostic marker in malignant mesothelioma with important roles in tumor growth [27]. In addition, mesothelioma patients have shown higher serum levels of angiopoietin-1 compared to patients with benign conditions [28].

Thrombospondin-1 was originally described as an angiogenesis inhibitor, while other data suggest that it could act as an angiogenesis stimulator by playing roles in endothelial cells adhesion, chemotaxis, and proliferation [29]. In this study, suppression of TSP-1 by SDC-1 over-expressing mesothelioma cells is in line with decreased angiogenesis, supporting the latter observation. TSP-1 is higher in pleural mesothelioma compared to normal pleura at mRNA level [30] and mesothelioma patients with higher levels of TSP-1 tend to have shorter survivals compared to patients with lower TSP-1 levels [31], suggesting a prognostic role for TSP-1. On the contrary, the tissue inhibitor of metalloproteinases-1 (TIMP-1) is acting as anti-angiogenic factor through inhibiting matrix metalloproteinases (MMPs) and also through blockage of endothelial cells migration [32]. Taken together, the alterations of angiopoietin-1, TSP-1, and TIMP-1 associated to SDC-1 overexpression collectively led to reduced proliferation, wound closure, and tube formation of HUVEC cells. In contrast, conditioned medium from SDC-1-silenced MM cells had no significant effect on the ability of HUVEC cells to form tubes, whereas, it significantly increased the ability of HUVEC cells to close wounds. Moreover, conditioned medium from SDC-1 overexpressing MM cells significantly inhibited the proliferation of HUVEC cells, whereas chemotactic migration of HUVECs was not affected. Wound healing involves both proliferation and migration; however, chemotaxis was not influenced in our system, thus it seems that inhibition of proliferation is the major driving force behind the inhibition of wound closure.

The pro-angiogenic factors HGF [33,34], NRG1β1 [30], and FGF4 [35,36] were, on the other hand, upregulated by the SDC-1 over-expression. The effect of these factors, however, seems to not have significant impact on HUVEC proliferation and migration in our system, probably due to a stronger synergistic effect of anti-angiogenic factors, or lack of activity despite their elevated level.

TGF-β1 has dual roles as inducer or inhibitor of cellular signaling, depending on its concentrations in the microenvironment, and presence or concentration of other angiogenic-related factors [37]. TGF-β1 in our system is upregulated by SDC-1 and it could also be responsible for the observed inhibitory effects of SDC-1 on endothelial cell tubule formation and proliferation, but a causal correlation remains to be further elucidated.

In order to disclose whether SDC-1 presence in supernatant is the key factor to impede the angiogenesis, we hampered its shedding by silencing one of the important enzymes in this process, MMP-7. Although other enzymes could also be involved in SDC-1 shedding, such as MMP-9 or heparanase, we found that silencing of MMP-7 by 45% decreased the SDC-1 shedding nearly 30%. This partial inhibition of SDC-1 shedding was sufficient to revert the inhibition of tube formation elicited by SDC-1 over-expression.

MMP-7 is required for cell migration and is quickly upregulated in response to epithelial injuries. Moreover, MMP7-mediated shedding of SDC-1 affects the activity of α2β1 integrins towards a lower affinity and therefore less cell-matrix interactions which facilitates the cell migration [38]. Whether lack of MMP-7 per se or lack of both MMP-7 and shed SDC-1 affect the migration and tube formation of endothelial cells remains to be tested.

As we know, angiogenesis is a process which is dependent on both stimulator and inhibitor factors and is regulated by a balance of these factors. This might describe that why not only pro- or anti- angiogenesis-related factors are affected by both SDC-1- overexpression or silenced. A greater understanding of regulation of angiogenesis by overexpression of SDC-1 needs further investigation.

Modulation of soluble angiogenic factors is crucial in cancer progression and angiogenesis. A recent study showed an increasing level of VEGF in the sera and pleural fluid of the exposed workers with and without clinical evidence of malignant pleural mesothelioma. This finding suggests an involvement of VEGF in pro-tumorigenic and pro-angiogenic activity in the malignant mesothelioma patients [39]. In the present study, analysis of pleural effusions showed that mesothelioma patients with higher levels of VEGF in pleural effusion had shorter survivals, consistent with other studies, suggesting that angiogenesis is a major factor and prognostic marker in mesothelioma [40,41]. Additionally, using the Proteome Profiler Array, we identified that other angiogenesis-related proteins including HGF, TGF-β1, FGF-4, NRG1-β1 are significantly upregulated in malignant mesothelioma cells, suggesting that these proteins may also have a prognostic and angiogenic role in MPM.

Shed SDC-1 level has been associated with cancer progression, chemotherapy resistance, and prognosis [24,42]. In myeloma cells, elevated levels of VEGF and shed SDC-1 activate endothelial cells [43]. One of the possible mechanisms for SDC-1 involvement in cancer is the shedding process. For instance, chemotherapy resistance in colorectal cancer has been attributed to EGFR phosphorylation and activation of the downstream pathway through growth factor binding ability of heparan sulfate chains existing on shed SDC-1 [42]. It has been shown that chemotherapy induces SDC1 shedding via increased levels of sheddases [44,45]. Such a mechanism is also known for increased angiogenesis in cancer by shed SDC-1 with binding to pro-angiogenic factors such as VEGF, FGF2, insulin-like growth factor 1 receptor—IGFR1 and presenting these growth factors to their respective receptors [44].

We have previously shown that mesothelioma patients with higher levels of shed SDC-1 had a shorter survival [19]. Now we show that this is associated with positive correlation between SDC-1 and VEGF. Taken together, these data suggest that combining measurements of both VEGF and shed SDC-1 in pleural effusions could be a useful prognostic indicator in malignant mesothelioma.

Collectively, our data demonstrate a potent anti-angiogenic effect of SDC-1 on HUVEC cells mediated by an array of soluble angiogenesis-related factors. To study these mechanisms, we have used one mesothelioma cell line where the expression of SDC-1 could be modulated. The generalization of these findings to mesotheliomas in general necessitates studies of additional cases, and the potential use of this anti-angiogenic effect for therapeutic and prognostic purposes remains to be further evaluated.

## 4. Materials and Methods

### 4.1. Cell Lines and Cell Culture Conditions

Human malignant mesothelioma cells (STAV-AB) were cultured in RPMI 1640 medium containing 1% L-glutamine and 25 mM HEPES buffer (Life Technologies Inc., Paisley, UK) supplemented with 10% human AB serum, in humidified atmosphere containing 5% (*v*/*v*) CO_2_ at 37 °C. These cells show epithelioid morphology and express low SDC-1 levels on the cell surface. SDC-1 was stably over-expressed by transfection with a plasmid vector carrying the human full-length SDC-1 gene. SDC-1 over-expressing cells were selected with geneticin (G418; Roche Diagnostics GmbH, Mannheim, Germany). Cells transfected with the same vector lacking SDC1 were used as negative control. The level of SDC-1 over-expression was ensured prior to all experiments by fluorescence activated cell sorting (FACS), ELISA and RT-PCR analyses (Appendix A). SDC-1 overexpression was defined as >1.5-fold increased SDC-1 to control ratio.

Human umbilical vein endothelial cells (HUVECs) were purchased from ATCC (LGC Standards, UK, cat.no. ATCC-CRL-1730) and were grown in F12K medium (Kaighn’s Modification of Ham’s F-12 Medium, cat.no.ATCC 30-2004). The complete medium constitute of F12K medium supplemented with 10% FBS (Gibco, Life Technologies, CA, USA), 1% penicillin-streptomycin (5000 units/mL, Gibco, Life Technologies, CA, USA), 1% fungizone (amphotericin B) (250 µg/mL, Gibco, Life Technologies, CA, USA), 0.1 mg/mL heparin (Sigma-Aldrich, MO, USA) and 0.05 mg/mL endothelial cell growth supplement (ECGS, Sigma-Aldrich, MO, USA).

### 4.2. SDC-1 Silencing by Transfection with Small Interfering RNAs (siRNAs)

In order to silence SDC-1, we used three different siRNAs (SDC-1 silencer pre-designed siRNA, s12632, s12633, and s12634) and silencer negative siRNA as a negative control. All three SDC-1 gene silencers and the negative control were from Ambion. The siRNAs and negative control were mixed with Lipofectamine RNAiMAX according to the manufacturer’s recommendation. Later, cells were transfected with siRNAs/Lipofectamine complexes and incubated at 37 °C for 6 h. In order to reduce cytotoxic effect of the reagents, medium was changed with fresh growth medium/serum after 6 h. Cells were incubated over-night and used for subsequent assays. Three independent experiments were performed. The level of SDC-1 expression was ensured prior to experiment by qPCR and Luminex (Appendix A).

Total RNA was isolated using the High Pure RNA Isolation Kit (Roche, Mannheim, Germany) according to the manufacturer’s instructions. The yield and purity of the RNA were estimated spectrophotometrically by measuring the UV absorbance at 260 and 280 nm with the NanoDrop spectrophotometer (NanoDrop Technologies Inc., Wilmington, DE, USA). cDNA synthesis was performed using First Strand cDNA Synthesis Kit (Amersham Pharmacia Biotech., Little Chalfont, Buckinghamshire, England).

The quantitative real time polymerase chain reaction (RT-PCR) was performed with Platinum^®^ SybrGreen qPCR SuperMix-UDG kit (Invitrogen) using DNA-polymerase with a set of sense/antisense primers (CyberGene AB, Sweden). The primers were the following sequences: SDC-1 fwd: TCT GAC AAC TTC TCC GGC TC and SDC-1 rev: CCA CTT CTG GCA GGA CTA CA and GAPDH fwd: ACA TCA TCC CTG CCT CTA CTG G and rev: AGT GGG TGT CGC TGT TGA AGT C. PCR conditions were an initial denaturation at 95°C, followed by 40 cycles of 95°C for 15 s and 62°C for 30 s. All reactions were performed in triplicates, using a total volume of 10 μL in an iCycler machine (BioRAD). The quantity of each target was normalized to GAPDH as reference gene.

### 4.3. Conditioned Medium Preparation

Equal numbers of SDC-1 over-expressing and mock control cells were seeded until a confluency of 70–80% was reached. Cells were starved over-night and second day medium was collected and kept on ice. Cells were trypsinized, spun down (400 G, 5 min), and counted. Same procedures were used for SDC-1 silenced and scramble control cells.

For HUVEC cell proliferation, tube formation, wound healing, and migration, conditioned medium was concentrated using ultrafiltration with Centriprep filters YM10 at 3000 RPM (2600× *g*) at 4 °C for 30 min (a volume of 10 mL conditioned medium was reduced to 1.5 mL). The volume was normalized to the cell number and RPMI was used to equalize the volume. Concentrated conditioned medium were stored at −80 °C until they were used.

### 4.4. Pleural Effusions

Thirty-nine pleural effusions from malignant mesothelioma patients were used to measure soluble SDC-1 and VEGF levels. Sixteen effusions were collected at the Department of Pathology and Cytology, Karolinska University Hospital in Huddinge, Sweden, between 2006 and 2012. Twenty-three malignant mesothelioma effusions were collected at the Chest Disease Department of Eskisehir Osmangazi University in Eskisehir, Turkey, between 2002 and 2004. The samples were collected before any treatment was given. The diagnoses were established according to the guidelines of cytological diagnosis of malignant effusions and verified by histopathology, immunohistochemistry, ultrastructural, or biomarker analyses. The study was approved by the ethical review board of Stockholm, Sweden (2009/1138-31/3), and the ethical review board of Eskisehir University, Turkey. The pleural effusion samples were centrifuged at 1700× *g* for 10 min and acellular supernatants were stored at −80 °C until further analysis.

### 4.5. Proteome Profiler Human Angiogenesis Antibody Array

Proteome Profiler Human Angiogenesis Antibody Array (R&D Systems, Inc., Minneapolis, MN, USA) was used to measure the relative expressions of angiogenesis-related proteins. Briefly, conditioned medium was collected from SDC-1 over-expressing, SDC-1 silenced cells, and their corresponding controls; i.e., EGFP and negative siRNA, respectively. The conditioned medium was mixed with a cocktail of biotinylated detection antibodies and added to membranes coated with primary antibodies against 55 angiogenesis-related proteins and incubated overnight at 4 °C. Then, membranes were incubated for 30 min with streptavidin-horseradish peroxidase (HRP) and chemiluminescence detection reagents were added for approximately 1 min. Dot blots were registered with CCD camera (FluorChem SP, Alpha Innotech, San Leandro, CA, USA). The average pixel density of duplicate spots on the membrane was determined using ImageJ software (http://rsb.info.nih.gov/ij/).

### 4.6. Endothelial Cell Proliferation Assay

HUVEC cells were seeded in triplicates into 96-well plates at a concentration of 2000 cell/well in complete HUVEC cell media. After 5 h complete medium was removed, and cells were starved overnight using the culture medium containing 1% FBS. The day after, starving medium was removed and conditioned medium from SDC-1 over-expressing or mock control cells containing recombinant human VEGF (10 ng/mL) (Peprotech, Rocky Hill, NJ, USA) was added to each sample. Conditioned medium from SDC-1 over-expressing or mock control cells was added to HUVEC complete medium in a 1:9 (vol/vol) ratio. HUVEC cell proliferation was assessed by Cell Proliferation Reagent WST-1 (Roche Diagnostics Scandinavia AB, Bromma, Sweden) at 24 and 48 h after addition of conditioned medium, according to the manufacturer’s instruction. Briefly, cells were incubated with 1/10 (*v*/*v*) WST1 reagent for 4 h at 37 °C. Optical densities were derived using a Spectramax spectrophotometer at 450 nm with background subtraction at 630 nm.

### 4.7. Wound Healing Assay

The potential of HUVEC cells to wound closure was assessed with wound healing assay. We used CELL BIOLABS CytoSelect 24-well wound healing assay kit. A wound was generated by readymade inserts in the wells, corresponding to standardized and highly reproducible the wound field. HUVEC cells (50,000 cells) were seeded in 24-well plate in 500 µL of HUVEC complete medium. Cells were incubated overnight at 37 °C to allow them to attach to wells. The day after, inserts were removed carefully from the wells and optimal medium was replaced with undiluted conditioned medium from SDC-1 over-expressing cells, SDC-1 silenced cells, and controls. Wound closure was observed at different time points (0, 4, 8 and 24 h) and captured at these time points with a digital camera connected to the inverted microscope (Eclipse TE100, Nikon, Tokyo, Japan). Images were analyzed and wound areas were measured with the ImageJ software. All experiments were repeated at least three times, and at least three images were captured at 10× magnifications along each wound. The slopes were calculated by the linear regression from zero to eight hours intervals to constrain the saturation due to cell growth.

### 4.8. Endothelial Cell Chemotaxis Assay

The chemotactic effect of soluble angiogenic factors released from SDC-1 over-expressing or mock control cells was evaluated using a modified two chamber assay with a pore size of 3 μm (Transwell chamber, Corning, NY, USA). For this experiment, 2.5 × 10^5^ HUVEC cells were seeded in culture medium containing 1% FBS in the upper chamber in duplicates. Conditioned medium from SDC-1 over-expressing cells or controls were added to the lower chamber. To initiate cell movement across the membrane, recombinant VEGF (10 ng/mL) was added to the conditioned medium. After 24 h incubation at 37 °C, the non-migrated cells in the upper chamber were carefully removed with a cotton swab, and the cells that passed through the membrane were fixed in 4% formaldehyde for 30 min at room temperature. Cells were stained with 0.2% crystal violet (Certistain, Merck, Germany) in 2% ethanol, and photographed randomly in four independent fields at 10× magnification for each well. The percentage of area covered by HUVEC cells in each image was analyzed using ImageJ software. The experiment was conducted three times.

### 4.9. Endothelial Cell Tube Formation Assay

In vitro VEGF-induced tube formation capacity of human umbilical vein endothelial cells (HUVEC) was assessed. HUVEC cells were incubated overnight with medium containing 1% FBS. On the second day, 50 μL of ECM gel solution was added to each well of a pre-chilled 96-well sterile plate and incubated for 30 min at 37°C. Then, 1.5 × 10^4^ cells per well were seeded in 3D culture on extra cellular matrix gel (ECM) (Cell Biolabs, San Diego, CA, USA) in triplicates in 96 well-plate coated with ECM gel. HUVEC cells were suspended in conditioned medium from SDC-1 over-expressing cells, SDC-1 silenced cells, and their controls with a ratio of 1:2 conditioned medium/HUVEC complete medium (vol/vol) containing recombinant human VEGF (10 ng/mL) (Peprotech, Rocky Hill, NJ, USA). Cells were incubated for 6 h at 37 °C in conditioned medium and then tube formation was monitored. Three images were digitally photographed at 10X magnification using a camera attached to a bright-field microscope (Eclipse TE100, Nikon, Tokyo, Japan). Images were analyzed using Wimasis image analysis software (Wimasis GmbH, Munich, Germany).

### 4.10. Interference with SDC-1 Shedding by MMP-7 Gene Silencing

In order to inhibit SDC-1 shedding, one of the important metalloproteinases involved in the shedding process, MMP-7, was silenced in STAV-AB cells over-expressing SDC-1 using small interfering RNA (siRNA) knockdown technique. Cells were transfected using Chem Cruz siRNA Reagent System (Santa Cruz Biotechnology, Inc., Dallas, TX, USA) one day after seeding (300,000 cells/well in a 6 well plate) with siRNA constructs specific for MMP-7 (Cat. No.sc-41553, Santa Cruz Biotechnology, Inc., Dallas, TX, USA) as it described previously (paragraph 4.2). Control cells were transfected with scrambled siRNA, with nonsense sequence. Three independent experiments were performed.

In order to check MMP7 silencing, mRNA levels were measured using RT-PCR. Total RNA, cDNA synthesis, and RT-PCR were performed as it described previously (paragraph 4.2). The MMP-7 primers were designed by us, using gene sequences from GeneBank (NCBI), with the following sequence: MMP7 fwd: GAG TGC CAG ATG TTG CAG AA and MMP7 rev: AAA TGC AGG GGG ATC TCT TT, and GAPDH fwd: ACA TCA TCC CTG CCT CTA CTG G and GAPDH rev: AGT GGG TGT CGC TGT TGA AGT C.

### 4.11. Soluble SDC-1 and VEGF Enzyme-Linked Immunosorbent Assay (ELISA)

SDC-1 and VEGF were measured using ELISA kits (human CD138; Gen-Probe Diaclone, France, cat. No. 950.640.192 and human VEGF; Quantikine ELISA, R&D Systems, Minneapolis, MN, USA, cat. No. DVE00, respectively). ELISAs were performed according to the manufacturers’ instructions. Effusions (for measuring SDC-1 and VEGF), cell supernatants, and cell lysates (for measuring SDC-1) were diluted to be fitted into exponential phase of standard curve, using kit-dilution buffers as blanks. Optical densities were determined spectrophotometrically (BioTek’sPowerWave HT, Winooski, VT, USA) at 450 nm. Pleural effusions were analyzed in duplicates, blinded to patient survival times.

### 4.12. SDC-1 Measurement by Luminex Assay with Human Premixed Multi-Analyte Kit

In order to measure SDC-1 expression level in SDC-1 silenced cells, Human Premixed Multi-Analyte kit from R&D system (LXSAHM-09) used to assess the level of syndecan-1. Samples were diluted 5-fold using dilution buffer from the kit. All standards and samples were assayed in duplicate. SDC-1 specific antibody is pre-coated onto magnetic microparticles embedded with fluorophores at set ratios for each unique microparticle region. Two spectrally distinct light emitting diodes (LEDs) illuminate the microparticles. One LED excites the dyes inside each microparticle to identify the region and the second LED excites the Streptavidin-PE to measure the amount of analyte bound to the microparticles. A minimum of 50 beads per region were counted. The median fluorescence intensities were determined on a Luminex 100/200 analyzer.

### 4.13. Fluorescence Activated Cell Sorting (FACS)

FACS was performed, in order to measure SDC-1 protein level in SDC-1 overexpressed cells. Syndecan-1 overexpressed cells and their respective controls were dissociated using 0.2% EDTA or scraped. Cells were stained with antibodies against SDC-1 (CD138) (1:10 dilution, Catalog no. MCA2459, Bio-Rad, Stockholm, Sweden) for 30 min on ice. The corresponding isotype IgG1 (1:10 dilution, Ref no. X0931, DAKO, Copenhagen, Denmark) was used as a negative control. Later, cells were incubated for 15 min at room temperature in dark with secondary antibody goat anti-mouse IgG Alexa Flour 488 (1:200, Catalog no. A-11017, Invitrogen, Stockholm, Sweden). Flow cytometry was performed using Becton Dickinson Flow Cytometry (Mountain View, CA, USA).

### 4.14. Statistical Analyses

Non-parametric Spearman test was computed for correlation analysis between soluble SDC-1 and VEGF in paired pleural effusions. Nonparametric tests were used since data were not normally distributed (D’Agostino and Pearson omnibus normality test; data not shown). The log-rank (Mantel–Cox) test compared survival curves and estimated hazard ratio and *p* values. Significance was analyzed using analysis of variance (ANOVA), two-tailed Student’s *t*-test, paired, or multiple *t*-test when needed. Significance was considered at *p* < 0.05.

To investigate if VEGF levels correlate with the survival of mesothelioma patients, survival analysis was performed. Survival times were available for a sub-set of patients. A cut-off value was used based on the highest and most significant hazard ratio using the online web application Cut-off Finder as previously described. Kaplan–Meier survival analysis was then used to check the statistical significance. The log-rank (Mantel–Cox) test compared survival curves and estimated hazard ratios and *p* values.

## 5. Conclusions

Our studies highlight the effects of syndecan-1 overexpression on growth factor secretion of mesothelioma cells which inhibits endothelial cells proliferation, tube formation, and migration. Additionally, we showed that VEGF could be used in prognostic evaluation of mesothelioma patients.

## Figures and Tables

**Figure 1 cancers-13-00655-f001:**
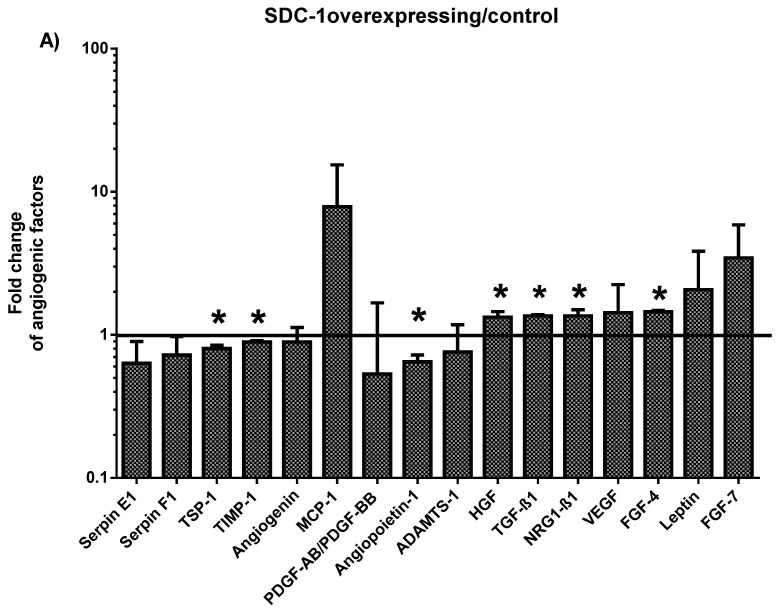
Effect of syndecan-1 (SDC-1) overexpression and silencing on the secretion of angiogenesis-related proteins in malignant mesothelioma. Relative level of angiogenesis-related proteins secreted to the conditioned medium by mesothelioma cells transfected with full length SDC-1/empty vector (**A**) and SDC-1 silenced/scrambled control cells (**B**). Angiogenic factors influenced by both SDC-1 over-expression and silencing are depicted relative to their respective specific controls. Black columns represent SDC-1 overexpression/empty vector control and gray columns represent SDC-1 silenced/scrambled control (**C**). The level of soluble angiogenic proteins was determined by Proteome Profiler™ Human Angiogenesis Antibody Array kit. Three independent experiments were performed. Values are represented as mean fold change. Error bars denote ± standard deviation (SD). Statistical significance was calculated using two-tailed Student’s *t*-test. Asterisks (*) indicate statistically significant differences (*p* < 0.05) compared to the corresponding control.

**Figure 2 cancers-13-00655-f002:**
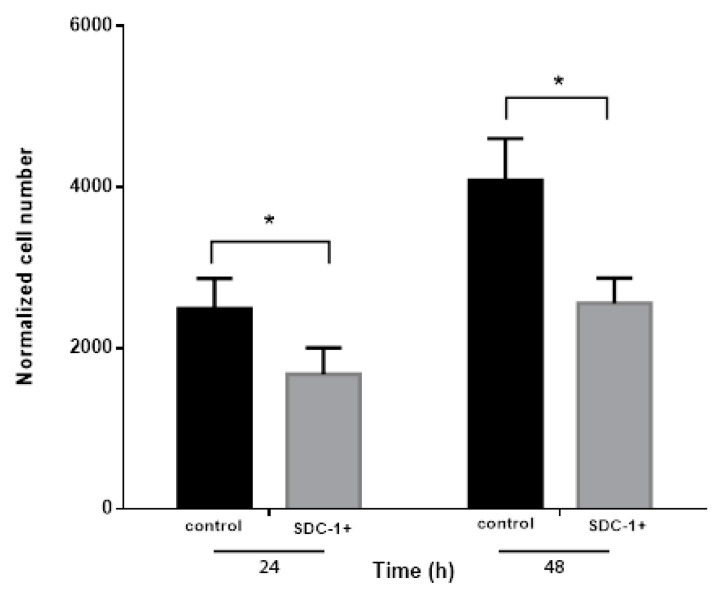
Syndecan-1 overexpression inhibits the proliferation of Human Umbilical Vein Endothelial Cells (HUVEC) cells. Proliferation of HUVEC cells is significantly inhibited after incubation with conditioned medium from SDC-1 over-expressing mesothelioma cells supplemented with 10 ng/mL Vascular Endothelial Growth Factor (VEGF) compared to the controls. Black bars denote HUVEC cells treated with control medium and gray bars show HUVEC cells treated with conditioned medium from SDC-1 over-expressing cells. Proliferation was measured using WST1 proliferation assay. Values represent mean cell number ± SEM, obtained from three independent experiments in triplicates. All cell numbers were normalized to the initial cell numbers after four hours incubation with conditioned medium. Paired *t*-test was performed to test the statistical significance. Asterisks (*) indicate statistically significant differences (* = *p* ≤ 0.05).

**Figure 3 cancers-13-00655-f003:**
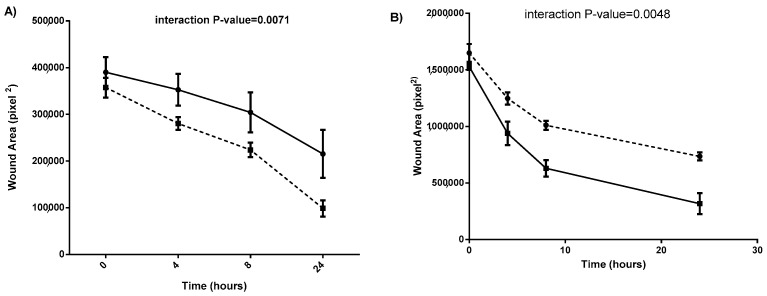
Overexpression of syndecan-1 SDC-1 inhibits the wound closure of endothelia cells, meanwhile SDC-1 silencing has the opposite effect. The closure of the wound area in HUVEC cells is hampered by the conditioned medium of SDC-1 over-expressing cells compared to the empty vector control (**A**) and it is accelerated by the conditioned medium from SDC-1 silenced cells compared to negative scrambled control (**B**). Black line denotes HUVEC cells treated with conditioned medium from SDC-1 over-expression or silencing, and dotted line shows HUVEC cells treated with control medium. Wound area was measured by ImageJ software. Mean values of three independent experiments performed in triplicates are shown. Error bars denote ± standard deviation (SD). Statistical significance was calculated using repeated two-way ANOVA (interaction *p* value < 0.05).

**Figure 4 cancers-13-00655-f004:**
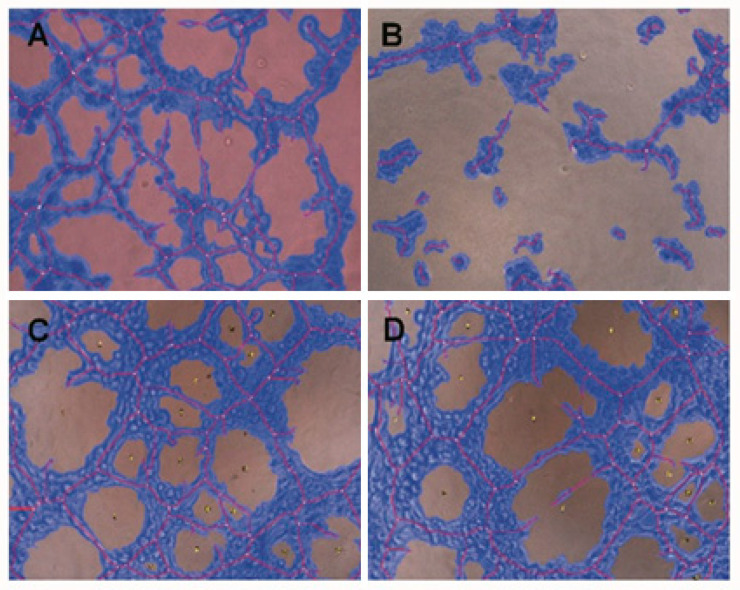
Capillary- like structures formed by HUVEC cells after incubation with conditioned medium from control cells (**A**); SDC-1 over-expressing cell (**B**), control medium from scrambled control cells (**C**), and MMP7 silencing cell medium (**D**). Images were taken with 10X magnification. Red line defines the tube’s length and blue color shows capillary structures. Small yellow dots demonstrate the branching points.

**Figure 5 cancers-13-00655-f005:**
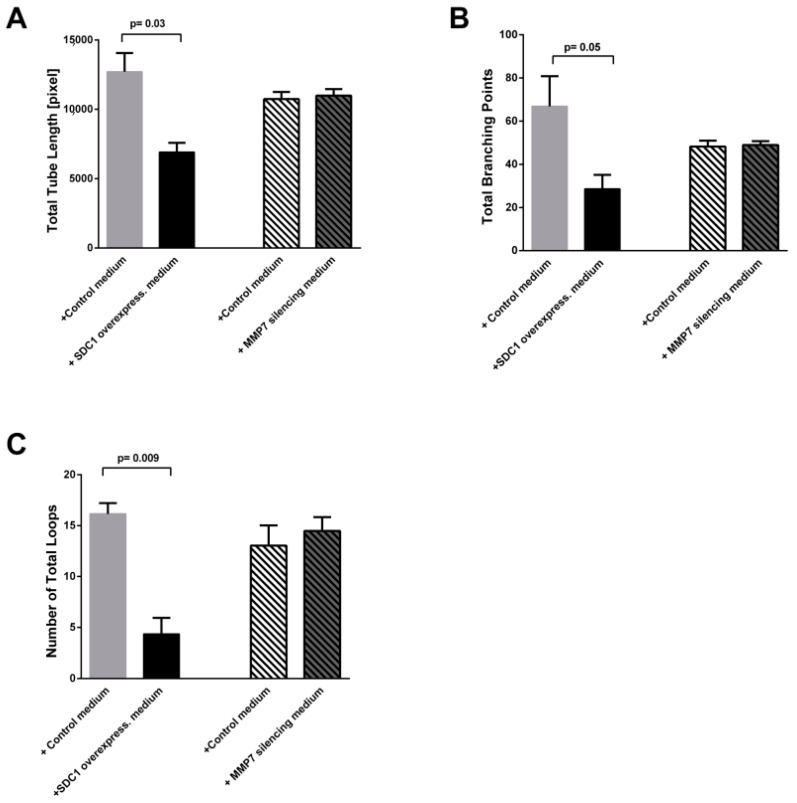
Tube formation of HUVEC cells. Conditioned medium from SDC-1 over-expressing malignant mesothelioma cells inhibit capillary-like tubule formation of HUVEC cells, while conditioned medium from MMP7 silencing cells does not affect tubule formation. (**A**) Total tube length, (**B**) Total branching points, (**C**) Number of total loops were significantly lower in HUVECs incubated with SDC-1 over-expressing cells conditioned medium compared to control. Paired *t*-test was performed to test the statistical significance, where *p* < 0.05. Three independent experiments were performed. Values correspond to mean ± SEM.

**Figure 6 cancers-13-00655-f006:**
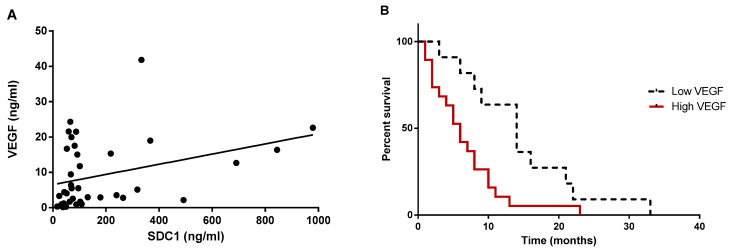
Prognostic value of SDC-1 and VEGF. Soluble VEGF correlates to soluble SDC-1 levels in pleural effusions and has prognostic roles in patients with malignant mesothelioma. (**A**) Spearman correlation analysis was used to assess the relationship between soluble VEGF and SDC-1 in paired sample effusions from 38 mesothelioma patients. (**B**) Cutoffs for “high” (n = 19) and “low” (n = 11) VEGF were identified by the online web application Cutoff Finder (2.125 ng/mL). The effect is presented in corresponding Kaplan–Meier plots. X axis is time in months and Y axis is survival percentage. *p* value is < 0.01 and it is based on log-rank (Mantel–Cox) tests.

## Data Availability

Data sharing not applicable.

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
