# Peer review of "Syndecan-1 Overexpressing Mesothelioma Cells Inhibit Proliferation, Wound Healing, and Tube Formation of Endothelial Cells"

_cancers, 2021, doi:10.3390/cancers13040655_

Round 1

Reviewer 1 Report

Authors studied the effect of conditioned media of mesothelioma (MM) cell with modified level of Syndecan-1 on proliferation, migration and tube formation of endothelial cells. Authors analyzed also samples derived from patients. Authors described that extracellular domain of SDC-1 have an impact on HUVEC cell behavior connected antigenic features.

Following are comments and concerns to improve this manuscript:

Broad comments:

Authors presented interesting data, however they are mostly rather descriptive without explaining the mechanisms of SDC-1 action on HUVEC cells. Moreover presented data are based only on conditioned medium cominf from culture one MM cell line with modified SDC-1. It would be better to perform experiments using several cell lines. Additionally not in all types of experiments authors present data for both types of manipulations (overexpression and silencing), thus the results seems to be not complete, what makes difficult to draw conclusions. Moreover some conclusions are weekly supported by obtained results.

Specific comments :

  • In the introduction authors describe the mechanism of activation and action of YAP, however it is difficult to understand, what is the connection between level of YAP and  SDC-1 or angiogenesis –related protein? If any exists?
  • Materials and methods is written sometimes too briefly and some methods seems to be described twice. For example:
    • Line 360-362: . “The level of SDC-1 over-expression was ensured prior to all experiments by fluorescence activated cell  sorting (FACS), ELISA and qPCR analyses (Supplementary figures 1B and 2B). “ – how RNA was isolated? What kind of primers (sequence)  or TaqMan probes? What was the FACS gating strategy? What is the supplier of ELISA?
    • Gene silencing is described twice: in 4.2 and 4.10 paragraph
    • Why in different types of experiments authors used not a constant dilution of MM conditioned media. For example for proliferation 1:9, for tube formation assay 1:2 and for wound healing is not specified?
  • At the figure 1 authors presented the level of angiogenesis related proteins secreted to the conditioned medium by syndecan-1 overexpressing and SDC-1 silenced cells. However it is difficult to compare figure A and B since not the same proteins are presented and they are in different order on graphs. Authors should simplify it. Moreover it would be nice to present also results from cells treated with recombinant extracellular domain of SDC-1, is it also changing the profile of secreted proteins? Moreover this type of control, where HUVEC cells are treated with HUVEC culture medium supplemented with recombinant SDC-1 should be included to all types of experiments performed (proliferation, migration, tubulogenesis assay) to verify if observed effect is connected with the level of SDC-1 in conditioned media or with modified profile of secretion of extracellular proteins.
  • Authors presented endothelial cell migration measured by wound healing assay. Pictures provided as supp Fig 4 are a poor quality, there is a problem with focus on them, moreover it seems that there is different magnification between fig A and B. Thus please provide a better quality photos. Despite this Authors presented that SDC-1 overexpression inhibits the proliferation of HUVEC. Therefore how authors prevent and ensured to analyze during wound healing assay migration and how they minimalize the effect of proliferation in this assay? And what is the effect of SDC-1 silenced cell conditioned medium on proliferation of HUVEC?
  • In the paragraph there are presented results from chemotaxis of HUVEC, but in my opinion they bring no new information to the work, so should be omitted or transferred to the supplementary.
  • Line 229: ”HUVEC cells incubated with medium from SDC-1 over-expressing cells exhibited slightly more cytoplasmic YAP” – based on provided microscopic data it is difficult to conclude that. It would be nice to present Western blotting analysis of YAP in nuclear and cytosolic fractions coming from HUVEC treated with SDC-1 overexpressing MM cell conditioned media. Moreover what is the phenotype of HUVEC cells incubated with SDC-1 silenced media?
  • According to Line 359 control and SDC-1 overexpressing cells express also GFP. Therefore how authors distinguished between GFP and alexa Fluor 488 fluorescence corresponding to YAP, since both are green? Next, authors performed analysis of cells directly cocultured - Line 492-495. How authors ensure in second part of fig 7 what kind of cell are analyzing HUVEC or MM, since they were seeded together on coverslips? Moreover single-cultured cells as a control are missing.
  • It seems that there is a mistake in scale bar in description of fig 7 legend: 20mm that is a 2 cm what makes that authors can observe cells without microscope.
  • Comments connected with discussion:
    • Line 281-281: ”TIMP-1 and TSP-1 were significantly upregulated upon SDC-1 silencing” , however on fig1B authors presented that only IL-8 level was changed in this conditioned medium.
    • Line 299-300: “In addition, we showed that silencing SDC-1 by 70% reverts the inhibition of tube formation” – there is some inaccuracy and misstatement since in line 202 authors wrote that there was no effect “Silencing of SDC-1 did not inhibit the formation of capillary-like structures of HUVEC cells”.

Author Response

Reviewer #1
Broad comments:
The study aims at showing possible effects of SDC-1 on HUVEC endothelial cells and angiogenesis. The tool to do so is a mesothelioma cell line, where the SDC1 expression had been modulated by transfection. This cell line is interesting, because SDC1 influences its differentiation along the epithelial-mesenchymal axis, so important for the prognosis of mesotheliomas. We completely agree that obtained findings can’t be directly generalized to all mesotheliomas, although presence of crucial factors were analyzed also in a large set of effusion supernatants from mesothelioma patients.
The basic expression of SDC-1 in this mesothelioma cell line is in the lower range, and the main experiment is to upregulate this. Silencing will only decrease the SDC1 expression from low to lower and is in this context not as informative.

The text has been modified to make this clear.

Specific comments:
1. In the introduction authors describe the mechanism of activation and action of YAP, however it is difficult to understand, what is the connection between level of YAP and SDC-1 or angiogenesis –related protein? If any exists?

The connection between SDC-1, TGFß, Hippo and YAP is now better presented in the text (lines 74-81).

2. Materials and methods is written sometimes too briefly, and some methods seems to be described twice. For example: Line 360-362: . “The level of SDC-1 over-expression was ensured prior to all experiments by fluorescence activated cell sorting (FACS), ELISA and qPCR analyses (Supplementary figures 1B and 2B). “ – how RNA was isolated? What kind of primers (sequence) or TaqMan probes? What was the FACS gating strategy? What is the supplier of ELISA?

The text has been expanded to describe this better (lines 362, 382-397, 488, and 491-492).

3. Gene silencing is described twice: in 4.2 and 4.10 paragraph

Silencing of SDC-1 is now only described in paragraph 4.2 and silencing of MMP-7 in paragraph 4.10.

4. Why in different types of experiments authors used not a constant dilution of MM conditioned media. For example, for proliferation 1:9, for tube formation assay 1:2 and for wound healing is not specified?

For the wound healing assay (paragraph 4.7) and the tube formation assay we used two different assay kits, as described. The respective protocols following these kits recommend different dilutions of conditioned media as described. In addition, dilution of conditioned media is not specified for wound healing because, HUVEC cells complete media was replaced completely with undiluted conditioned media from SDC-1 overexpressed and silenced cells.

5. At the figure 1 authors presented the level of angiogenesis related proteins secreted to the conditioned medium by syndecan-1 overexpressing and SDC-1 silenced cells. However, it is difficult to compare figure A and B since not the same proteins are presented and they are in different order on graphs. Authors should simplify it. Moreover, it would be nice to present also results from cells treated with recombinant extracellular domain of SDC-1, is it also changing the profile of secreted proteins? Moreover, this type of control, where HUVEC cells are treated with HUVEC culture medium supplemented with recombinant SDC-1 should be included to all types of experiments performed (proliferation, migration, tubulogenesis assay) to verify if observed effect is connected with the level of SDC-1 in conditioned media or with modified profile of secretion of extracellular proteins.

The use of recombinant/artificial or isolated SDC-1 is a way of studying the specific effect of this PG. Such a design would, however, not show the complete picture of effects from increased SDC-1 expression, i.e., the effects of shed SDC-1 together with all other simultaneously secreted products, whose expression also is influenced by the altered SDC-1. The present design better reflects the biological situation, where tumor cells influences the behavior of neighboring endothelium.

The figure is changed as requested.

6. Authors presented endothelial cell migration measured by wound healing assay. Pictures provided as supp Fig 4 are a poor quality, there is a problem with focus on them, moreover it seems that there is different magnification between fig A and B. Thus, please provide a better quality photos. Despite this, authors presented that SDC-1 overexpression inhibits the proliferation of HUVEC. Therefore, how authors prevent and ensured to analyze during wound healing assay migration and how they minimalize the effect of proliferation in this assay? And what is the effect of SDC-1 silenced cell conditioned medium on proliferation of HUVEC?

We agree that the supplementary Figure 4 is of insufficient quality (out of focus and uneven light). Furthermore, all information regarding the results is given in the text. We therefore delete the figure and all references to it in the text.
As regards the second part of this comment, we are not certain about what the reviewer asks for. The measure obtained in the “wound healing assay” was that of decreased wound healing when exposed to media from cells overexpressing SDC-1. This measure is a combination of proliferation and migration. To elucidate this further we analyzed both cell migration in a two-chamber system and proliferation. The altered wound healing was mainly associated with altered proliferation. Further silencing the low expression of SDC-1 had no effect. This is now better explained in. the text (lines 301-303).

7. In the paragraph there are presented results from chemotaxis of HUVEC, but in my opinion they bring no new information to the work, so should be omitted or transferred to the supplementary.

These measurements are a part of understanding the wound healing experiment. See previous point.

8. Line 229: ”HUVEC cells incubated with medium from SDC-1 over-expressing cells exhibited slightly more cytoplasmic YAP” – based on provided microscopic data it is difficult to conclude that. It would be nice to present Western blotting analysis of YAP in nuclear and cytosolic fractions coming from HUVEC treated with SDC-1 overexpressing MM cell conditioned media. Moreover what is the phenotype of HUVEC cells incubated with SDC-1 silenced media?

We agree that Western Blot would be a nice way to quantify the presence of YAP. However, the use of combined cultures is necessary to study this. Western Blot is therefore not an alternative, since proteins from the two cell lines can’t be isolated separately. So, the evaluation must rely on morphology, in this case confocal microscopy. The two cell types are easily distinguished since they have quite different nuclear:cytoplasmic ratios and also nuclear size.

The quantification is now included in Figure 7.

9. According to Line 359 control and SDC-1 overexpressing cells express also GFP. Therefore, how authors distinguished between GFP and alexa Fluor 488 fluorescence corresponding to YAP, since both are green? Next, authors performed analysis of cells directly cocultured - Line 492-495. How authors ensure in second part of fig 7 what kind of cell are analyzing HUVEC or MM, since they were seeded together on coverslips? Moreover single-cultured cells as a control are missing.

This was a mistake fby us. We have not at all used the EGFP transfectants in these experiments. The text is corrected accordingly (paragraph 4.1)

10. It seems that there is a mistake in scale bar in description of fig 7 legend: 20mm that is a 2 cm what makes that authors can observe cells without microscope.

This is now corrected (20 µm instead of 20 mm, line 240).

11. Line 281-281: ”TIMP-1 and TSP-1 were significantly upregulated upon SDC-1 silencing” , however on fig1B authors presented that only IL-8 level was changed in this conditioned medium.

This is now better described in lines 280-284.

12. Line 299-300: “In addition, we showed that silencing SDC-1 by 70% reverts the inhibition of tube formation” – there is some inaccuracy and misstatement since in line 202 authors wrote that there was no effect “Silencing of SDC-1 did not inhibit the formation of capillary-like structures of HUVEC cells”

This is now better described in lines 301-303

Reviewer 2 Report

The authors studied SDC-1 expression on mesothelioma cells, its intersaction with angiogenesis factors and its ability to inhibit proliferation, migration, as well as tubule formation of endothelial cells. They found that upregulation of SDC-1 expression resulted in inhibition of endothelial cells, while downregulation had the opposite effects. These findings are interesting and could help develop better angiogenesis inhibitor therapy. The manuscript is well written and documented. 

Minor comments:

Introduction:

Para 1. While the majority of pleural malignant mesotheliomas do occur in patients with exposure to asbestos, there is a subset of cases where exposure cannot be documented. Furthermore, a larger proportion of peritoneal malignant mesotheliomas occur in patients lacking a history of asbestos exposure.

Page 2 , line 78: please specify what the NF2 product is that interacts (i.e. protein, RNA etc):

“MST1/2, Lats1/2 and Hippo, by neurofibromatosis type 2 (NF2) leads to phosphorylation”

Results:

Figure 2. The authors may wish to show what the different bars represent in the figure itself not just in the figure legend.

Author Response

Reviewer #2
Specific comments:

1. While the majority of pleural malignant mesotheliomas do occur in patients with exposure to asbestos, there is a subset of cases where exposure cannot be documented. Furthermore, a larger proportion of peritoneal malignant mesotheliomas occur in patients lacking a history of asbestos exposure.

The text has been changed accordingly (lines 50-51)

2. Please specify what the NF2 product is that interacts (i.e. protein, RNA etc):

This is now specified (a suppressor protein, line 77)

3. Figure 2. The authors may wish to show what the different bars represent in the figure itself not just in the figure legend.

The figure has been improved as suggested

Round 2

Reviewer 1 Report

Authors modified only in the minor way the manuscript, did not provide necessary controls or experiments, what means that they did not respond to addressed questions and comments in the acceptable way. Please find below specific comments:

  1. At the figure 1 authors presented the level of angiogenesis related proteins secreted to the conditioned medium by syndecan-1 overexpressing and SDC-1 silenced cells. However, it is difficult to compare figure A and B since not the same proteins are presented and they are in different order on graphs. Authors should simplify it. Moreover, it would be nice to present also results from cells treated with recombinant extracellular domain of SDC-1, is it also changing the profile of secreted proteins? Moreover, this type of control, where HUVEC cells are treated with HUVEC culture medium supplemented with recombinant SDC-1 should be included to all types of experiments performed (proliferation, migration, tubulogenesis assay) to verify if observed effect is connected with the level of SDC-1 in conditioned media or with modified profile of secretion of extracellular proteins.

    The use of recombinant/artificial or isolated SDC-1 is a way of studying the specific effect of this PG. Such a design would, however, not show the complete picture of effects from increased SDC-1 expression, i.e., the effects of shed SDC-1 together with all other simultaneously secreted products, whose expression also is influenced by the altered SDC-1. The present design better reflects the biological situation, where tumor cells influences the behavior of neighboring endothelium.

    Despite I agree with authors, that the use of recombinant/artificial or isolated SDC-1 is a way of studying the specific effect of this PG, but in my opinion it is necessary. It should serve as a control to presented results and to demonstrate the specific effect of secreted proteins’ level modified after SDC- 1 overexpression. Since after silencing of SDC-1 authors did not observed spectacular changes in secretion profile of angiogenic factors and as it was written in manuscript “HUVEC cell migration (…) is accelerated by the conditioned medium from SDC-1 silenced cells” it suggests that SDC-1 is a key player in observed effect.  Additionally I would not agree with the authors, that “Silencing will only decrease the SDC1 expression from low to lower and is in this context not as informative” since the conditioned media from SDC-1 cells affects biological functions of HUVES it indicates, that it is rather necessary to complete the results to have comprehensive panel of information about SDC-1 possible action. 

  1. Authors presented endothelial cell migration measured by wound healing assay. Pictures provided as supp Fig 4 are a poor quality, there is a problem with focus on them, moreover it seems that there is different magnification between fig A and B. Thus, please provide a better quality photos. Despite this, authors presented that SDC-1 overexpression inhibits the proliferation of HUVEC. Therefore, how authors prevent and ensured to analyze during wound healing assay migration and how they minimalize the effect of proliferation in this assay? And what is the effect of SDC-1 silenced cell conditioned medium on proliferation of HUVEC?

    We agree that the supplementary Figure 4 is of insufficient quality (out of focus and uneven light). Furthermore, all information regarding the results is given in the text. We therefore delete the figure and all references to it in the text.
    As regards the second part of this comment, we are not certain about what the reviewer asks for. The measure obtained in the “wound healing assay” was that of decreased wound healing when exposed to media from cells overexpressing SDC-1. This measure is a combination of proliferation and migration. To elucidate this further we analyzed both cell migration in a two-chamber system and proliferation. The altered wound healing was mainly associated with altered proliferation. Further silencing the low expression of SDC-1 had no effect. This is now better explained in. the text (lines 301-303).

Since authors can not distinguish between proliferation and migration in wound healing assay they should not call it “migration assay”, because as they admit, both processes were estimated. Moreover, explanation “To elucidate this further we analyzed both cell migration in a two-chamber system and proliferation” is not convincing, since authors presented at fig.4 that  (line 185-186)“. Chemotactic migration of HUVEC cells is not affected upon incubation with conditioned media from SDC-1 over-expressing cells when compared to mock control cells.” It means that media from SDC-1 over-expressing cells affect proliferation not migration as authors conclude, and altered wound healing was mainly associated with altered proliferation. It should be modified and explained in the manuscript. Moreover conclusions about impact of SDC-1 overexpression on HUVEC cell migration (line 560-561, 44-45, 272-273 etc) should be supported by other more believable results (authors may use for example mitomycin c in would healing assay to ensure that they are focusing on cell migration or apply other cell migration assay, like single cell tracking). Without other results this conclusion should be omitted in the manuscript.

  1. Line 229: ”HUVEC cells incubated with medium from SDC-1 over-expressing cells exhibited slightly more cytoplasmic YAP” – based on provided microscopic data it is difficult to conclude that. It would be nice to present Western blotting analysis of YAP in nuclear and cytosolic fractions coming from HUVEC treated with SDC-1 overexpressing MM cell conditioned media. Moreover what is the phenotype of HUVEC cells incubated with SDC-1 silenced media?

    We agree that Western Blot would be a nice way to quantify the presence of YAP. However, the use of combined cultures is necessary to study this. Western Blot is therefore not an alternative, since proteins from the two cell lines can’t be isolated separately. So, the evaluation must rely on morphology, in this case confocal microscopy. The two cell types are easily distinguished since they have quite different nuclear:cytoplasmic ratios and also nuclear size.

Authors presented results ”HUVEC cells incubated with medium from SDC-1 over-expressing cells exhibited slightly more cytoplasmic YAP”(line 229-230), thus in this experimental settings single cell type – HUVEC was treated with conditioned medium collected from cancer cells, so there is a possibility to isolate protein extract only from 1 cell line and in this case  it seems that Western Blot from in nuclear and cytosolic fraction is rather not a complicated experimental operation. Moreover I agree that the use of combined cultures where cell are cultured in one dish is the best way to study interactions between 2 cell types. However there exist several ways to do Western Blot after direct culture of two different types of cells. Cells can be sorted (FACS-sorted or MACS-separated) based on different cell surface markers (HUVEC cells should be positive for CD31 (PECAM), CD144 (VE-cadherin), CD146). Therefore, there is the method to separate two cell lines and next isolate different cellular fractions. Moreover this method of distinguish two cell lines based on different nuclear:cytoplasmic ratios and also nuclear size (not described in manuscript or in other reference) is not convincing.

4.Line 281-281: ”TIMP-1 and TSP-1 were significantly upregulated upon SDC-1 silencing” , however on fig1B authors presented that only IL-8 level was changed in this conditioned medium.

This is now better described in lines 280-284.

 In the revised version of manuscript authors wrote: “In mesothelioma cells, among the angiogenesis related proteins Angiopoietin-1, thrombospondin-1 (TSP-1) and tissue inhibitor of metalloproteinases-1 (TIMP-1) were significantly downregulated upon SDC-1 over-expression, whereas IL-8 was upregulated upon SDC-1 silencing. In addition, among proteins which were altered by both SDC-1 over-expression and silencing, TIMP- 1 and TSP-1 were significantly upregulated upon SDC-1 silencing.”

It means exactly the same as in 1st version of manuscript. According to my knowledge all data needs to go through statistical tests (analysis). The results of these analysis indicate if some variable is significantly up- or down-regulated, and this significance is presented on graphs for example as asterisk(s) above some bars. If statistical test dos not indicate p- value below 0,05 we can not conclude about significant changes. On fig1B authors presented that only IL-8 level was changed significantly in this conditioned medium -there is asterisk only above 1 bar, thus authors can not claim that TIMP- 1 and TSP-1 were significantly upregulated upon SDC-1 silencing, since it is not supported by the presented results.

5.Line 299-300: “In addition, we showed that silencing SDC-1 by 70% reverts the inhibition of tube formation” – there is some inaccuracy and misstatement since in line 202 authors wrote that there was no effect “Silencing of SDC-1 did not inhibit the formation of capillary-like structures of HUVEC cells”

This is now better described in lines 301-303

 In the revised version of manuscript authors wrote:” In addition, we showed that silencing SDC-1 by 70% did not inhibit the tube formation and increase the migration of HUVEC cells. This leads to the conclusion that SDC-1 seems to be crucial for angiogenesis inhibition.”

This phrase should be modified. Authors did not observed any effect of silencing of SCD-1 on tube formation and it should be clearly explained. Moreover since authors admit that observed would healing closure is rather connected with cell proliferation this should be also written in the manuscript to avoid confusions.

Author Response

Dear Madam/Sir,

Please find enclosed our revised manuscript where we addressed the constructive criticism, that helped us to improve our manuscript:

  1. We have now provided additional information for our statements regarding SDC-1 expression in malignant mesothelioma. Appropriate controls are better explained for each set of experiment and the rationale of the experimental design and experimental procedure is better clarified now. The second revision is marked in blue in the main manuscript and in supplementary figures.

Our study aims to study how Syndecan1 (SDC-1) synthesis influences the secretion of 55 angiogenesis related proteins and thereby endothelial cells proliferation, wound closure and tube formation. Our tool is a malignant mesothelioma cell line, where we’ve modulated the SDC1 expression by transfection and silencing.  From a range of broad molecular profiling studies that we performed over the years we know that SDC-1 expression is low in this mesenchymal tumor. Therefore; our major approach is SDC-1 overexpression, whereby we have previously shown that tumor proliferation can be inhibited and the mesenchymal phenotype can be reverted to an epithelioid phenotype, finally resulting in slower tumor progression in SCID mice. In contrast to overexpression SDC-1 silencing elicits only modest or no effects on gene expression profiles and most biological assays.

In the current study we show that SDC-1 overexpression significantly alters the soluble agiogenesis related proteins that collectively lead to profound inhibition of tube formation of HUVEC endothelial cells along with inhibition of their proliferation. No such an effect was seen by SDC-1 silencing or MMP-7 silencing. MMP-7 act as a main sheddase that responds to large extent to the removal of the SDC-1 ectodomain from the cell surface to the cell culture medium, thereby converting it to a soluble molecule. By silencing both SDC-1 and its main sheddase MMP-7 we have shown that not SDC-1 ectodomain alone, but together with other excreted factors mediates its inhibitory effect on endothelial tube formation.

The use of recombinant SDC-1 protein would be biologically less relevant compared to our current experimental approach. The recombinant protein is most likely not modified post-translationally, thus it will lack glycosylation and sulphation; features that are crucial for the complex biological effects exhibited by SDC-1. Principally, most known SDC-1 effects are mediated through these interactions with a complex pattern of other biologically active molecules.

  1. We changed the text regarding the wound healing assay and its relevance in our experimental system.

  1. To mimic the biological situation where tumor cells and endothelial cells are interacting directly and through paracrine effector molecules when they grow side by side in a tumor, we’ve used co-cultures of the two cell lines (HUVEC endothelial cells and mesothelioma cells).

HUVEC cells in single culture are our control cells, where we could not observe changes in the subcellular localization of YAP1 protein. The subcellular distribution of YAP1 is controlled by attachment to the ECM and other cells, thus it can be best studied attached to a surface and with high resolution confocal microscopy. Any manipulation or detachment from the ECM will cause rearrangements in YAP1, thus FACS or Western Blotting would profoundly alter the YAP1 distribution and would not show thru attachment mediated subcellular distribution.

We have described in materials and methods the criteria for differentiating between HUVEC and tumor cells morphologically. The evaluation was performed with confocal microscopy that allows a high resolution and detailed subcellular examination and in situ quantitation of the nuclear and cytoplasmic fractions of YAP. The morphology of the two cell lines differ a lot (HUVEC with large cytoplasm and small nucleus while the meso cell line have larger nuclei and much less cytoplasm, there are no “intermediate” morphologies).

  1. The Figure 1 C shows the significant changes that are present in both SDC-1 overexpressing and silenced cells.

  1. Changes are done in agreement with the suggestions.

Sincerely Yours:

Katalin Dobra

Professor/Senior Consultant in Molecular Pathology

Karolinska Institutet/Karolinska University Hospital

Stockholm, Sweden

Round 3

Reviewer 1 Report

Authors modified only in the minor way the manuscript, did not provide necessary controls or experiments, what means that they did not respond to addressed questions and comments in the acceptable way.

  • On the Fig 1C author compare two different types of genetic manipulations with each other what in my opinion is inappropriate and give a conclusions which ale false.
  • Authors wrote” HUVEC cells in single culture are our control cells, where we could not observe changes in the subcellular localization of YAP1 protein. The subcellular distribution of YAP1 is controlled by attachment to the ECM and other cells, thus it can be best studied attached to a surface and with high resolution confocal microscopy. Any manipulation or detachment from the ECM will cause rearrangements in YAP1, thus FACS or Western Blotting would profoundly alter the YAP1 distribution and would not show thru attachment mediated subcellular distribution.”

In this situation additional marker of HUVEC/cancer cells should be included in the immunofluorescent staining to distinguish two cell type in confocal microscopy. Despite this authors based on the method which is not convincing and has no support in the scientific literature. Moreover despite the fact that the quantification of YAP by WB is probably possible (like in article: Lee, Y., Kim, N.H., Cho, E.S. et al. Dishevelled has a YAP nuclear export function in a tumor suppressor context-dependent manner. Nat Commun 9, 2301 (2018).) authors should at least provide the quantification of the nuclear/cytoplasmic ratio of YAP based on confocal pictures in the middle confocal slice of the nucleus (like A Das et al.  J Biol Chem (2016) PMID: 26757814, PMCID: PMC4813550, DOI: 10.1074/jbc.M115.708313).

  • There is no explanation in the manuscript that in this case wound healing is mostly driven by proliferation. Authors omitted this fact.
  • In the previous version of manuscript authors wrote:” In addition, we showed that silencing SDC-1 by 70% did not inhibit the tube formation and increase the migration of HUVEC cells. This leads to the conclusion that SDC-1 seems to be crucial for angiogenesis inhibition.”

This phrase should be modified. Authors did not observed any effect of silencing of SCD-1 on tube formation and it should be clearly explained. Moreover since authors admit that observed would healing closure is rather connected with cell proliferation this should be also written in the manuscript to avoid confusions. Authors assured that “Changes were done in agreement with the suggestions” , however no modifications were done in discussion part.

Author Response

Response to reviewer round 3

Thanks for the reviewer for the constructive criticism that helped us to improve our paper.

Please find below the point by point response to the comments:

  • On the Fig 1C author compare two different types of genetic manipulations with each other what in my opinion is inappropriate and give a conclusions which ale false.

      Response: This is a misunderstanding. The presented data do not correspond to direct comparison. We compare the effect of the conditioned medium secreted by SDC-1 overexpressing cells/conditioned medium of empty control and calculate the ratio (black column) and similarly calculate the ratio of SDC-1 silenced/ versus control conditioned medium (grey column). Thus, in the graph we show normalized ratios to their corresponding controls. We emphasized this in the figure 1c and the corresponding legend text.

  • Authors should at least provide the quantification of the nuclear/cytoplasmic ratio of YAP based on confocal pictures in the middle confocal slice of the nucleus (like A Das et al.  J Biol Chem (2016) PMID: 26757814, PMCID: PMC4813550, DOI: 10.1074/jbc.M115.708313).

       Response: We quantified the nuclear YAP according to the reviewer’s suggestion; we made a new figure 6 and changed the legend text and text accordingly. We agree with the reviewer that the co-culture data are difficult to interpret and did not add to the main context, so we excluded it, further highlighting the effect of conditioned medium of the SDC-1 overexpressing MM cells.

  • There is no explanation in the manuscript that in this case wound healing is mostly driven by proliferation. Authors omitted this fact.

       Response: Included now this in the text. Lines: 298-304

  • In the previous version of manuscript authors wrote:” In addition, we showed that silencing SDC-1 by 70% did not inhibit the tube formation and increase the migration of HUVEC cells. This leads to the conclusion that SDC-1 seems to be crucial for angiogenesis inhibition.” This statement has to be modified.

Moreover since authors admit that observed would healing closure is rather connected with cell proliferation this should be also written in the manuscript to avoid confusions. Authors assured that “Changes were done in agreement with the suggestions”, however no modifications were done in discussion part. 

Response: We explain this better in the text. Lines: 298-304